# Genome-Wide Analysis of the *UGT* Gene Family and Identification of Flavonoids in *Broussonetia papyrifera*

**DOI:** 10.3390/molecules26113449

**Published:** 2021-06-06

**Authors:** Fenfen Wang, Yalei Su, Naizhi Chen, Shihua Shen

**Affiliations:** 1Key Laboratory of Plant Resources, Institute of Botany, The Chinese Academy of Sciences, Beijing 100093, China; wangfenfen@ibcas.ac.cn (F.W.); suyalei@ibcas.ac.cn (Y.S.); 2University of Chinese Academy of Sciences, Beijing 100049, China

**Keywords:** *Broussonetia*, polyphenols, flavonoid, antioxidant activity, *BpUGTs*

## Abstract

*Broussonetia papyrifera* is a multifunctional deciduous tree that is both a food and a source of traditional Chinese medicine for both humans and animals. Further analysis of the UGT gene family is of great significance to the utilization of *B. papyrifera*. The substrates of plant *UGT* genes include highly diverse and complex chemicals, such as flavonoids and terpenes. In order to deepen our understanding of this family, a comprehensive analysis was performed. Phylogenetic analysis showed that 155 *BpUGTs* were divided into 15 subgroups. A conserved motif analysis showed that *BpUGT* proteins in the same subgroups possessed similar motif structures. Tandem duplication was the primary driving force for the expansion of the *BpUGT* gene family. The global promoter analysis indicated that they were associated with complex hormone regulatory networks and the stress response, as well as the synthesis of secondary metabolites. The expression pattern analysis showed that the expression level of *BpUGTs* in leaves and roots was higher than that in fruits and stems. Next, we determined the composition and content of flavonoids, the main products of the *BpUGT* reaction. A total of 19 compounds were isolated and analyzed by UPLC-ESI-MS/MS in 3 species of *Broussonetia* including *B. kazinoki*, *B. papyrifera*, and *B. kazinoki × B. papyrifera*, and the number of compounds was different in these 3 species. The total flavonoid content and antioxidant capacities of the three species were analyzed respectively. All assays exhibited the same trend: the hybrid paper mulberry showed a higher total flavonoid content, a higher total phenol content and higher antioxidant activity than the other two species. Overall, our study provides valuable information for understanding the function of *BpUGTs* in the biosynthesis of flavonoids.

## 1. Introduction

Flavonoids, as a type of polyphenolic compound with a C6-C3-C6 double aromatic ring, have been extracted from the leaves, skins, roots and fruits of many plants [1]. Modern pharmacological studies have shown that the flavonoids are beneficial in drug development and health care, such as anti-cancer [2], anti-oxidation [3], anti-inflammation [4], anti-bacteria [5], anti-allergy [6], anti-tumor [7] and other pharmacological activities. In plants, flavonoids exist in various modified forms, and are generated by hydroxylation, methylation, acylation, and glycosylation, among which glycosylated flavonoids are the most common natural compounds. Most of the natural flavonoids are *C*-glycosides and *O*-glycosides, and the most abundant flavonoid glycosides in plants are flavone glycosides [8]. The most reported flavonoid *O*-glycosides are 7- and 3-*O*-glycosides, and the flavonoid *C*-glycosides are found mainly as 6- and 8-*C*-glycosides. Although the flavonoid *C*-glycosides are less well known than flavonoid *O*-glycosides, they exhibit a wide range of benefits for human health [9]. The glycosylation is mainly catalyzed by glycosyltransferases (GTs), which are classified into 111 families (http://www.cazy.org/GlycosylTransferases.html) (accessed on 5 January 2021). Both flavonoid *C*-glycosides and *O*-glycosides are catalyzed by uridine diphosphate (UDP)-sugar dependent glycosyltransferases (UGTs), which belong to GT family-1. The sequences at the N-terminal region of these enzymes are highly diverse and are considered to be responsible for the recognition of a variety of substrates, and the C-terminal region contains a conserved motif called Plant Secondary Product Glycosyltransferase (PSPG). The PSPG box is a unique, well conserved region of 44 amino acids which has been found in all the *UGTs* of all studied plants [10]. In addition to flavonoids, these enzymes also act on other substrates in plants, such as terpenoids and auxin, to regulate plant growth, development, disease resistance, and the interaction with the environment [11].

*Broussonetia* genus, which belongs to the Moraceae family, distributes naturally in Asia and Pacific countries. There are four species including *B. papyrifera*, *B. kazinoki*, *B. kaempferi*, and *B. kurzii* [12]. In China, *B. papyrifera* (paper mulberry) has been used as a traditional Chinese medicine to treat bleeding, dropsy, and dysentery disease and has a long history [13]. Four bioactive compounds including phenolic compounds, terpenoids, flavonoids and alkaloids have been isolated from *B. papyrifera* [14]. In particular, the ethanol extracts from *B. papyrifera* have received extensive attention due to their inhibitory effects on cancer cells [15]. Uralenol and broussochalcone A are flavonoid compounds from *B. papyrifera*, which have shown inhibitory activities against the PTP1B (Protein Tyrosine Phosphatase 1 B) enzyme [16]. Meanwhile, prenylflavone derivatives from *B. papyrifera* have also shown an inhibitory effect on the growth of breast cancer cells in vitro and in vivo [17]. Additionally, the fruit extract could also be used in dietary supplement preparations as a food additive and to prevent the oxidation of food products [18]. A hybrid paper mulberry (*B. kazinoki* × *B. papyrifera*) was obtained by employing crossbreeding biotechnology [19], which has a great adaptability to climates and soils, a rapid growth rate, multi-resistance to pests and diseases, and a high protein content in the leaves. Meanwhile, it has been widely used in ecological restoration, paper making, medicine and livestock [20]. The leaves of the hybrid paper mulberry have been used as a kind of non-conventional forage feed, which is superior to compound feed because of its ability to reduce the need for antibiotics and its ability to improve the meat quality. It is worth mentioning that the hybrid paper mulberry, as a prospective woody species, has been added in the feed raw materials catalogue in 2018 (http://www.moa.gov.cn/govpublic/XMYS/201810/t201810246161429.htm) (accessed on 13 June 2020). Consequently, it is necessary to accelerate the research focus on the comprehensive utilization of the hybrid paper mulberry.

In this study, the *UGT* genes were identified using the sequenced *B. papyrifera* genome and a systematic analysis was performed, including the phylogenetic relationships, gene structure, conversed motifs, chromosomal location, duplication event, *cis*-regulatory analysis and expression patterns in different tissues. Through these analyses, we hoped to identify all the *UGT* genes in paper mulberry and to understand the function of *BpUGTs* in the biosynthesis of flavonoids. Flavonoids were qualitatively and quantitatively analyzed in leaf samples from three *Broussonetia* species by UPLC-ESI-MS/MS (ultra-high performance liquid chromatography mass spectrometry coupled to a triple-quadrupole mass spectrometry with electrospray ionization). Nineteen metabolites were isolated and identified in total, including phenols and flavones. Our results showed that the composition and content of secondary metabolites in the leaves from three species were different. Among these species, the content of the main bioactive metabolites in the leave extracts of the hybrid paper mulberry was higher than that in *B. kazinoki* and *B. papyrifera*, so the hybrid paper mulberry can be used as a potential germplasm resource for extracting specific drugs in the future.

## 2. Materials and Methods

### 2.1. Plant Materials

The leaves were collected randomly from plants in mid-July 2018 (39°48′ N, 116°28′ E, 76 m) in the Institute of Botany, the Chinese Academy of Sciences. The chosen experimental plants had been grown for more than 2 years in an identical growth environment. All leaves, which were located in the same place, were rapidly frozen in liquid nitrogen and were dried with a vacuum centrifuge concentrator (CV100-DNA, Baijiu, Beijing, China). The dried experimental materials were stored at −20 °C before analysis. All concentrations used in this study were calculated by dry weight (DW). Three independent biological replicates were performed for each plant.

### 2.2. Chemical and Reagents 

Standard neochlorogenic acid and chlorogenic acid were purchased from Fisher Chemicals (Shanghai, China). The flavone *C*-glycosides standards of luteolin, luteolin-8-*C*-β-D-glucopyranoside (orientin), luteolin-6-*C*-β-D-glucopyranoside (isoorientin), luteolin-7-*O*-β-D-glucopyranoside, apigenin, apigenin 8-*C*-β-D-glucopyranoside (vitexin), and apigenin 6-*C*-β-D-glucopyranoside (isovitexin) were obtained from Alltech Scientific (Beijing, China). The purity of all standards was HPLC grade. Acetonitrile, methanol, and formic acid used for UPLC-MS were of chromatographic grade. The hydrochloric acid and the acetic acid of analytical grade were purchased from Sigma-Aldrich (Shanghai, China). Double distilled water was produced by using a Milli-Q System (Millipore, Billerica, MA, USA). 1,1-diphenyl-2-picrylhydrazyl (DPPH), 2,2′-azino-bis (3-ethylbenzothiazoline-6-sulphonic acid) (ABTS) and 2,4,6-tripyridyl-S-triazine (TPTZ) were purchased from Sigma-Aldrich (St. Louis, MO, USA). The Millipore membrane filters (D = 0.22 μm) were purchased from ANPEL Scientific Instrument Corporation (Shanghai, China).

### 2.3. Identification of UGT Family Genes in Paper Mulberry Genome

To identify the *UGT* family genes in the paper mulberry genome, two approaches were used. Firstly, the known 122 UGT protein sequences of *A. thaliana* were downloaded from the TAIR database v10.0 (https://www.arabidopsis.org/) (accessed on 5 January 2021) and were used as queries to search the UGT protein database by using a local BLASTP program [21]. Secondly, the Hidden Markov Model (HMM) seed file of the UGT domain (Pfam00201, http://pfam.xfam.org/) (accessed on 5 January 2021) was also used to search the paper mulberry candidate proteins. Then, each of these HMM models were used as a probe to perform a BLASTP against the local paper mulberry protein sequence database by using HMMER 3.0 (http://www.hmmer.org/) (accessed on 5 January 2021). Summarizing the results of both methods and removing the redundant sequences, the remaining sequences were the candidate UGT protein sequences of paper mulberry. The candidate protein sequences were further verified by scanning against SMART (http://smart.embl-heidelberg.de/) (accessed on 7 January 2021), PFAM, and CDD Search (https://www.ncbi.nlm.nih.gov/Structure/cdd/wrpsb.cgi) (accessed on 7 January 2021) to confirm the presence of the UGT domain in each candidate sequence.

### 2.4. Phylogenetic, Gene Structure and Conserved Motif Analysis

For phylogenetic analysis, the full-length amino acid sequences of UGT proteins from *Arabidopsis* and *B. papyrifera* were aligned by MEGA X software with default parameters. An unrooted phylogenetic tree was constructed by the maximum likelihood (ML) method based on the JTT amino acid substitution model with 1,000 bootstrap replicates. The iTOL (http://itol.embl.de/help.cgi) (accessed on 10 January 2021) online tool was used to better illustrate and edit the phylogenetic tree. The gene structure of the *BpUGT* genes were examined by the online website GSDS 2.0 (http://gsds.cbi. pku.edu.cn) (accessed on 13 January 2021). The MEME Suite web server 5.1.1 (Multiple Expectation Maximization for Motif Elicitation) (http://meme-suite.org/tools/meme) (accessed on 16 January 2021) was used to analyze the conserved motifs of the *BpUGT* sequences with default arguments, the number of repetitions was set to 0 or 1, and the maximum number of motifs was set to 20.

### 2.5. Chromosomal Location, Gene Duplication Events and Promoter Cis-Regulatory Analysis

The chromosomal location information of all the *BpUGT* genes were extracted from the paper mulberry genome database and were visualized by the MapInspect version 1.0 (https://mapinspect.software.informer.com/) (accessed on 3 February 2021) graphical tool. For the identification of gene duplication, all the *BpUGT* amino acid sequences were searched against the paper mulberry genome by employing the BLASTP process with an e-value of 1 × 10^−5^. To identify the *cis*-elements in the promoter sequences of the *UGT* family genes in paper mulberry, 2,000 bp genomic sequences upstream of the transcription start site were analyzed using the PlantCARE database (http://bioinformatics.psb.ugent.be/webtools/plantcare/html/) (accessed on 3 February 2021).

### 2.6. Expression Analysis of BpUGT Genes in Different Tissues

The expression profiles of *BpUGTs* genes in ten tissues (fruit, shoot apex, young leaf, developing leaf, mature leaf, immature stem, phloem of proximal stem, phloem of mature stem, phloem of root, and root tip) were detected using transcriptome data. Each tissue had three biological replicates. The transcript abundance was represented by fragments per kilobase of the transcript per million fragments mapped (FPKM) values which were calculated based on RNA-seq reads. The expression values of each gene in the different tissues were averaged and presented as a log_2_ value. The results were presented as heatmaps using the TBTOOLS software.

### 2.7. Sample Extraction

The extraction of the flavonoids was performed according to the method described by Chen et al. [22] with minor modifications. The lyophilized leaves powder (20 mg) was transferred into 2 mL centrifuge tubes containing 1 mL of methanol-acetic acid (0.1%), the mixtures were shaken with a vortex-qilinbeinbeier (Kylin-Bell, Haimen, China) for 30 s, the samples were sonicated by a KQ-50B ultrasonic cleaner (Ultrasonic instruments, Kunshan, China) at room temperature for 1 h and then centrifuged in SIGMA 2K15 (Sigma Centrifuges, Germany) at 12,000 rpm for 10 min. The supernatants were collected in new 2 mL centrifuge tubes. The sample extraction was filtrated by millipore membrane filters (D: 0.22 μm) prior to UPLC-DAD and UPLC-MS^2^ analysis.

### 2.8. UPLC-PDA and UPLC-MS/MS System and Conditions

The samples were examined with ultra-performance liquid chromatography (UPLC, Waters, Milford, MA, USA) coupled to a triple-Quadrupole Mass Spectrometry (XEVO^®^-TQ) with electrospray ionization (ESI). The separation was carried out with a ZORBAX Eclipse plus C18 (150 mm × 3.0 mm) with a particle size of 1.8 µm (Agilent Technologies, Santa Clara, CA, USA) at 40°C. The gradient was calculated using 0.1% formic acid (A) and acetonitrile (B) as the mobile phases, 0–1 min (5% B), 1–8 min (5–40% B), 8–12 min (40–95% B), 16–17 min (95–100% B), 17–21 min (100% B), 21–25 min (5% B). The operating conditions were set at positive ion ESI modes with a 1.0 mL/min flow rate. Chromatograms were acquired at 350 nm and photodiode array spectra were recorded from 200 to 800 nm. The UPLC-MS/MS analysis for flavonoids was performed using a XEVO^®^ TQ-MS triple quadrupole mass spectrometer (Waters, Milford, MA, USA), which was connected to an Ultra Performance Liquid Chromatograph (UPLC-MS/MS, Waters). The UPLC separation conditions were the same as mentioned above. The flavonoids were employed in a positive ion (PI) mode and the MS detection conditions were as follows: capillary voltage, 3.00 kV for PI mode; cone voltage, 30V for PI mode; desolvation gas flow, 650 L/h; cone gas flow, 50 L/h; collision gas flow, 0.12 mL/min; collision energy, 15 eV for PI mode; desolvation temperature, 300 °C; source temperature, 150 °C; scan range, 100–1000 (*m*/*z*). The injection volume of each sample was 1 µL. Analytical software (MassLynx™, V 4.1, SCN 846, Waters Corp., Manchester, UK) was used for the system control and the data processing. For data acquisition, each MS scan and UV detection were acquired three times.

### 2.9. Quantitative and Qualitative Analysis of Flavonoids

The compounds were identified by comparing them with the retention times of the standards, the characteristics of the UV-vis spectra of the peaks and the mass spectrometric information with Mass Hunter qualitative software. Quercetin 3-rutinoside (rutin) and trans-5-caffeoylquinic acid were used as the standards for the semi-quantification of all flavonoids and chlorogenic acids, respectively. The linear regression equations are as follows: for rutin, Y = 0.0041X + 0.0032 (r^2^ = 0.9991); for chlorogenic acid, Y = 0.0021X + 0.0058 (r^2^ = 0.9982). The concentrations of phenolic compounds were presented as micrograms of corresponding standards per 1 g of DW. All analyses were performed with three biological replicates.

### 2.10. Total Polyphenol and Flavonoid Contents 

The total number of polyphenol compounds was determined by means of the Folin–Ciocalteau reaction as previously reported [23]. A sample solution (100 μL, 5 mg/mL) was added to 100 μL of the Folin–Ciocalteau reagent and then mixed with 1000 μL of double distilled water and 200 μL of 20% sodium carbonate. The reaction solution was incubated at room temperature for 1 h and the absorbance of the sample mixture was recorded at 760 nm. To obtain the total phenolics contents, the calibration curve (Y = 2.853X + 0.0072, r^2^ = 0.9986) established that the gallic acid (GA) was used as standard. The results were expressed as mg gallic acid equivalents (GAE)/100 g dry weight (mg GAE/100 g DW). To measure the flavonoids in the leaf extracts, we used the spectrophotometer method as reported [23]. The reaction system included a 100 μL sample, 1000 μL of double distilled water and 500 μL of AlCl_3_ (2%, *v*/*v*) reagent. The sample absorbance was read at 430 nm after the mixture was stored for 15 min at room temperature. To obtain the value of the flavonoid compounds, the calibration curve (Y = 1.1748X + 0.0066, r^2^ = 0.9993) was established that the rutin was used as a reference and the results were expressed as mg rutin equivalents (RE)/100 g dry weight (mg RE/100 g DW).

### 2.11. Antioxidant Capacity Analysis

The antioxidant capacity was determined using free radical 1,1-diphenyl-2-picrylhydrazyl (DPPH), the kits of the ferric reducing ability of plasma (FRAP), and 2, 2’-azino-3-ethylbenzothiazoline-6-sulfonic acid (ABTS) assays. The specific operations were as follows. The DPPH assay was determined according to the reported protocols [24,25]. DPPH (0.2 mM) was dissolved in methanol. The sample (100 μL) was mixed with DPPH (2900 μL) and incubated in the dark at room temperature. After 30 min, the absorbance was measured at 515 nm. GA was used as an authentic standard and the calibration curve was established by plotting the DPPH·scavenging ratio (A_0_ − A_t_)/A_0_ (A_0_ and A_t_ separately mean the absorbance at the initiation and termination of the reaction) against the GA concentration. The linear regression equation was Y (scavenging ratio) = 13.554X + 0.0184 (r^2^ = 0.9992). The antioxidant capacity was expressed as GAE and data were presented as milligrams of GAE per 100 g DW. 

The FRAP assay was performed based on the method of previous studies [23]. A total of 60 mL of 10 mM of TPTZ solution (dissolved in 40 mM of hydrochloric acid), 60 mL of 20 mM of ferric chloride solution, and 600 mL of an acetate buffer solution (pH 3.6) were used to prepare the FRAP working solution. The sample (100 μL) was mixed with 2900 μL of the FRAP working solution to react in the dark at room temperature for 15 min. Then the absorbance at 593 nm was measured. Similarly, gallic acid (GA) was used as a standard and the calibration curve was Y (absorbance) = 25.522X − 0.0158 (r^2^ = 0.9999). Moreover, data were presented as milligrams of GAE per 100 g DW.

The ABTS assay was carried out based on the method described by the authors in [26]. The FRAP working solution contained 5 mL of 7 mM of ABTS solution, 1 mL of 140 mM of potassium persulfate solution, and 380 mL of the ethanol buffer solution. The sample extract and GA (100 μL) were added into 2900 μL of the ABTS working solution to incubate at room temperature. After 6 min, the absorbance at 414 nm was recorded using a UV spectrophotometer. GA was used as an authentic standard and the calibration curve was Y (absorbance) = 24.483X − 0.0059 (r^2^ = 0.9995). GAE was used to describe the results.

### 2.12. Statistical Analysis

Data was obtained from three independent biological replicates and the significance was determined via one-way analysis of variance (ANOVA), and the significant difference was employed when *p* < 0.05.

## 3. Results

### 3.1. Identification and Phylogenetic Analysis of BpUGTs

A total of 155 putative *BpUGTs* were identified in the paper mulberry genome using the HMM and BLASTP programs. The length of the deduced UGT proteins varied from 238 to 874 amino acids, with an average of 475. The predicted molecular weight ranged from 26 to 97 kDa. The isoelectric point ranged from 5.03 to 8.7 (Appendix A). The phylogenetic analysis of the identified *UGTs* was performed to analysis their grouping pattern and their genetic relationships based on the Arabidopsis *UGT* sequences. The paper mulberry *UGTs* were classified into 15 subgroups (A-O), including one newly discovered group (group O), which was present in *B. papyrifera* but not in *A. thaliana* (Figure 1).

The number of *UGTs* in each group varied: the largest group E had 46 *UGT* members and the smallest group N had only one member. The new group identified in our study was O containing two *UGT* members. There were 46 *UGT* members in the largest E group and only one *UGT* member in the smallest N group. 

### 3.2. Chromosome Location, Gene Duplication Events, Gene Structure and Conserved Motif Analysis of Paper Mulberry UGTs

To detect detailed information of the *UGT* genes in paper mulberry, we investigated the chromosome location of the *UGT* genes in the *B. papyrifera* species according to the gene annotation files retrieved from public genomic databases. After curation, 145 *UGT* genes in *B. papyrifera* were located on chromosomes, which represent 93.5% of the total *UGT* genes in paper mulberry (Figure 2).

The remaining 10 were mapped to the scaffold. In the *B. papyrifera* genome, the chromosome containing the greatest number of *UGT* genes (22 members) were the chr04 and chr09 chromosomes, and the chr05 chromosome only contained two *UGT* genes, and contained the least number of *UGT* genes compared with the other chromosomes. The chromosome location of the *UGT* genes is uneven in paper mulberry genomes. The *UGT* genes of group E with the most members (46 genes), were randomly distributed across 9 chromosomes (chromosome 1 and 3–10) and the remaining three members were located on the scaffold. The distribution of the *UGT* genes on each chromosome was also uneven but most were clustered, suggesting that there may be a gene duplication in the evolutionary process. Gene duplications were considered to play an important role in the expansion and evolution of the gene family, so a duplication event analysis of the *BpUGT* genes was performed. The results indicated that 155 *BpUGTs* were involved in 7 segment duplication events and 28 tandem duplication events, which suggested that tandem duplications may be the primary driving force for the expansion of the *BpUGT* family (Appendix A, Appendix A). The phylogenetic groups G, E, L, F, A and H possessed the maximum number of tandem duplicated *UGTs* (8, 5, 5, 3, 2 and 2, respectively), while J, M and I possessed one only. Groups B, D, K, N and O did not have any tandem duplicated *UGTs*. To better explore the relationships between the structure and the function of paper mulberry *UGT* genes, and to further clarify the evolutionary relationships within the *UGT* gene family, the exon/intron structure was analyzed. Among the 155 *BpUGTs*, 59 had no introns, and 63 contained only one intron (Appendix A). Within the same subgroup, most members shared similar exon/intron numbers and arrangements, and they had important sequence characteristics, indicating that they had very close evolutionary relationships (Appendix A). Conserved motifs occupy an important role in the characteristic analysis and classification of a gene family. The results showed that a total of 20 motifs were identified among the BpUGT proteins (Appendix A). 

### 3.3. Cis-Regulatory Analysis on Promoters and Expression Analysis of BpUGT Genes

*Cis*-acting regulatory elements play an important role in the regulation of gene transcription initiation through interactions with their corresponding trans-regulatory factors, especially in the synthesis of secondary metabolites. To obtain more valuable information, we analyzed the promoter regions of each putative *BpUGT* gene. Our results showed that these identified *cis*-regulatory elements could be classified into four functional categories: light response, hormone response, development regulation and stress response. Overall, these findings demonstrated that the *UGT* gene family in paper mulberry play a vital part in the complex hormone regulatory network and may be involved in a variety of stress responses, as well as the synthesis of secondary metabolites, which was helpful to explore the regulatory mechanisms of the family of *BpUGTs* (Appendix A). To obtain a broader understanding of the potential functions of *BpUGTs*, their expression patterns were analyzed using RNA-seq data of ten tissues (fruit, shoot apex, young leaf, developing leaf, mature leaf, immature stem, phloem of proximal stem, phloem of mature stem, phloem of root and root tip). FPKM values were used to evaluate the gene expression level. *BpUGT* genes exhibited different expression levels in different tissues. A total of 65 genes showed low expression levels; 80 genes showed high expression levels. The other 10 genes were not detected among all the examined tissues. Most of the highly expressed *BpUGT* genes were expressed more in the leaves and roots than in the fruits and stems (Figure 3, Appendix A).

Additionally, many of the *BpUGT* genes showed the highest level of transcript in leaves, indicating that UGTs may play an important role in the biosynthesis of glycosylated secondary metabolites.

### 3.4. Qualitative Analysis of Flavonoid Compounds

The *UGT* genes catalyze the glycosylation of most flavonoids. To identify the flavonoids and their derivatives, we employed UPLC-Q-TOF-MS in positive ion modes to analyze the leaf extracts, and the compounds were obtained under excellent chromatographic condition with good peak separation and resolution (Figure 4).

By combining UV absorption maxima, retention time and mass spectra, a total of 19 compounds were definitely or tentatively identified from the leaf samples of *Broussonetia*. The number of isolated compounds from *B. papyrifera*, *B. kazinoki* and hybrid paper mulberry were 18, 13 and 19, respectively. Different compounds are shown among the studied samples in Table 1. The 19 compounds were identified based on the UPLC-ESI-MS/MS analyses and by comparison with data with those standards or in the literature. 

All extracts were analyzed in positive ion mode (*m*/*z*, [M + H]^+^) using UPLC-ESI-MS/MS. These compounds could be divided into two main groups: phenolic acids and flavonoids (apigenin derivatives and luteolin derivatives). The first and second compounds (peaks one, two) showed the same [M + H]^+^ parent ion at the value *m*/*z* 355 and the maximum absorbance at 325 nm on the UV spectra. According to the basis of these characteristics, they were considered as the isomers of chlorogenic acid [27]. Thus peaks one and two were separately identified as neochlorogenic acid and chlorogenic acid by co-chromatography with corresponding standards. According to the UPLC-ESI-MS/MS analyses and comparison with standards and literature data, the flavonoid was considered to be a derivative of two flavones, apigenin and luteolin. Usually, the glycosylated flavonoids connect the sugars, mainly pentose (arabinose and xylose) and hexose (glucose, galactose and rhamnose) [28]. Thus, there are two kinds of connections between sugars and flavonoids when they are glycosylated, namely, *O-C* and *C-C* connections [29]. The former is relatively common, while the latter generally occurs in specific plant groups, and most of the glycosides are located in C6 and/or C8 positions [30]. Combined with molecular ions at *m*/*z* 287 [M]^+^ and 271 [M]^+^ in PI mode, the peaks 18 and 19 were regarded as luteolin and apigenin through UV spectra and by co-eluting with their counterpart standards. Peaks 8 and 10 were tentatively assigned as apigenin isomers, namely 6-*C*-pentosyl-8-*C*-glucosyl apigenin or 6-*C*-glucosyl-8-*C*-pentosyl apigenin because they showed the same fragment ions at *m*/*z* 565 [M + H]^+^ and 433 [M + H]^+^. There is no doubt that peaks 8 and 10 were identified as 6-*C*-glucosyl-8-*C*-arabinosyl apigenin (schaftoside) and 6-*C*-arabinosyl-8-*C*-glucosyl apigenin (isoschaftoside) by co-eluting with their corresponding standards under the same conditions. Mass spectrometry data from the isomers of 9 and 12 showed that there were two substituents: one molecule of glucose and one rhamnose linked to the aglycone apigenin. A previous study has demonstrated that 6-*C*-hexosyl isomers are eluted earlier than the 8-*C*-hexosyl isomers [24]. So, peaks 9 and 12 were characterized as 6-*C*-rhamnosyl-8-*C*-glucosyl apigenin (Ap-6-*C*-Rha-8-*C*-Glc) and 6-*C*-glucosyl-8-*C*-rhamnosyl apigenin (Ap-6-*C*-Glc-8-*C*-Rha), which had been reported in the *B. papyrifera* leaves [31]. Peaks seven and five were characterized as luteolin hexosides because they exhibited the same UV absorption wavelength and parent ions. Finally, peaks 7, 5 and 11 were separately identified as luteolin 8-*C*-β-D-glucopyranoside (orientin), luteolin 6-*C*-β-D-glucopyranoside (isoorientin), apigenin 8-*C*-β-D-glucopyranoside (vitexin), and they were further confirmed by co-elution with their corresponding standards. Orientin, isoorientin, and vitexin were previously separated from mulberry leaves [31]. Five flavonoid *O*-glycosides were identified by the characteristics of UV-vis absorption spectroscopy and means of co-elution with an authentic standard. With the parent ion at *m*/*z* 449 [M + H]^+^ and aglycone ion at *m*/*z* 287 [M + H−162]^+^ in PI mode, the fragment at *m*/*z* 287 corresponded to the [M + H − 162] ^+^ ion, indicating the presence of a glucose moiety. There was no doubt that peak 13 was luteolin-7-*O*-β-D-glucopyranoside by co-elution with the standard. In positive ionization mode, peak 14 showed a parent ion at *m*/*z* 463 [M + H]^+^ and a fragment ion at *m*/*z* 287, which gave similar mass spectrum data as peak 13. The fragment at *m*/*z* 287 corresponded to the [M + H − 176]^+^ ion, indicating the presence of a glucuronic acid moiety, thus peak 14 was tentatively identified as luteolin-7-*O*-β-D-glucopyranuronide. Peaks 15 and 16 were characterized as apigenin derivatives due to the aglycone ion at *m*/*z* 271 and the UV spectrum data. MS fragmentation data (loss of *m*/*z* 162) for peak 15 suggested that it was linked with a hexose, so peak 15 was confirmed as apigenin-7-*O*-β-D-glucopyranoside and peak 16 was characterized as apigenin-7-*O*-β-D-glucopyranuronide. Peak six was tentatively identified as vitexin-7-*O*-β-D-glucopyranoside by comparing the MS spectrum and UV data with the literature values. These five flavonoids had been previously described in mulberry leaves [31]. These flavonoid *O*-glycosides have various biological properties, such as antimicrobial, antifungal, and anti-inflammation properties [32]. Peak 3 produced pseudo-molecular ions at *m*/*z* 611 [M + H]^+^, peak 4 produced pseudo-molecular ions at *m*/*z* 595 [M + H]^+^ and peak 17 produced pseudo-molecular ions at *m*/*z* 477 [M + H]^+^ in PI mode. So, peaks 3, 4 and 17 were tentatively assigned as Di-*C*, *C*-hexosyl-luteolin, Di-*C*, *C*-apigenin and Dihydro-apigenin derivatives. 

### 3.5. Quantification of Flavonoids and Analysis Antioxidant Capacity in Sample Leaves

In order to further quantify the content of flavonoids, we carried out three quantitative analyses of the flavonoid content in the *Broussonetia* leaves under the same experimental conditions (Appendix A). In hybrid paper mulberry, the content of chlorogenic acid and neochlorogenic acid were the highest among the 19 chemical compounds analyzed. Regarding flavone, the luteolin content of the sample leaves was 27.4 ± 2.4 μg/g in hybrid paper mulberry, while the content was 2.6 ± 0.1 and 3.4 ± 0.4 μg/g in *B. kazinoki* and *B. papyrifera*, respectively. Furthermore, the apigenin content was 9.2 ± 0.6 μg/g in hybrid paper mulberry, which was the highest among the three *Broussonetia* species leaves and the apigenin content was 2.2 ± 0.2 μg/g in *B. kazinoki*, but the apigenin was not found in *B. papyrifera*. These results suggested that the hybrid paper mulberry leaves were more suitable for the production of healthcare products and were useful for the development of breeding programs. Taking both the composition and content into consideration, the hybrid paper mulberry could serve as another important natural source of flavonoid *C*- or *O*-glycosides. The total phenolic and flavonoid contents in the leaf extracts were analyzed as shown in Figure 5. 

Regarding the flavonoid content, it showed a significantly higher value in hybrid paper mulberry (5381.9 mg RE/100 g) than other leaves (3663.6 in *B. papyrifera* and 1830.7 mg RE/100 g in *B. kazinoki*, respectively). For total phenols, the content in the hybrid paper mulberry leaf extracts was the highest with 2909.2 mg GAE/100 g DW. In *B. papyrifera* and *B. kazinoki* extracts, the content of the total phenols was 1928.5 mg GAE/100 g and 1167.9 mg GAE/100 g, respectively. Phenols are the main natural antioxidants in plants, and a previous study also suggested that the higher total phenolic compound values reflected higher antioxidant activities. The antioxidant activity determinations (DPPH, FRAP and ABTS assays) of *Broussonetia* leaves are presented in Table 2. 

Our results demonstrated that the hybrid paper mulberry leaves possessed higher antioxidant abilities than those of *B. papyrifera* and *B. kazinoki* leaves. Finally, the hybrid paper mulberry could be considered as a new source for natural antioxidants. To confirm the effective components, we should further study the anti-tumor, anti-oxidation, and antimicrobial activity of total flavonoids from hybrid paper mulberry leaves based on the previous research.

## 4. Discussion

### 4.1. Potential Functions of BpUGT Genes Inferred from the Expression Patterns

Glycosylation is one of the most important modification and detoxification phenomenon in plant secondary metabolites, which is mediated by a set of GTs. GTs can be classified into at least 111 families, of which, the *UGT* gene family is the largest one [33]. UGTs have been identified and analyzed in a few plant species such as Arabidopsis [34], wheat [35], and cotton [10]. To date, there is no systematical analysis of the *UGT* gene family of paper mulberry. Therefore, in order to deepen our understanding of this family, a comprehensive analysis of phylogenetic relationships, gene duplication, gene location, conserved motifs, intron/exon position, and gene expression was performed. Phylogenetic analysis defined 15 distinct phylogenetic groups in paper mulberry, providing a useful foundation for the understanding of the structure-function relationships among the *UGT* family members. Our result showed that the largest group E consisted of sixteen 71-family *UGTs*, nine 72-family *UGTs*, six 88-family *UGTs*, and fifteen 80-family *UGTs*. In addition, many plant *UGT* gene members belonging to group E have been functionally identified, including the glycosylation of small molecule volatile compounds, and the synthesis of flavonoid glycosides and anthocyanins [36,37], which indicates that group E made an important contribution to the glycosylation of plant secondary metabolites. The tandem duplication event performs an important function in the expansion of gene families. After curation, we obtained 28 tandem-duplicated genes belonging to the members of the 9 *UGT* gene families representing 18.06% of the total *UGT* genes in paper mulberry. These results suggest that the tandem duplication event occurs continuously in the *UGT* gene family and is an ongoing process throughout the evolutionary history of the *UGT* gene family. The gene structure and the conserved motif analysis will be useful for further understanding of the *UGT* genes in paper mulberry. The intron analysis suggested that the conserved intron changed during the evolution of paper mulberry. The analysis of promoter regions suggested that some of the *UGT* genes contain a secondary metabolite related element, including MBSI *cis*-regulatory elements, implying that nine *UGT* genes might play a significant role in the synthesis of flavonoids. The expression analysis of *UGTs* provided candidates for a further function study of *UGT* genes in regulating flavonoid development. The expression analysis in this study gives us a global landscape of the expression of paper mulberry *UGTs* in different tissues. A more detailed experiment is still needed to determine the mechanism of *BpUGTs*’ at flavonoid initiation and the following steps of flavonoid development. Our study provided systematical insights into the potential roles of *UGTs* in paper mulberry, which is helpful for screening candidate genes and studying the functions of *UGT* genes, but a series of experiments are still required to confirm their functions in the future.

### 4.2. Overview of Polyphenol Compounds in Paper Mulberry

There are many flavonoids that are the main components in the compound preparation of traditional Chinese medicine. Therefore, it is not only of theoretical significance, but also of great practical value to study the flavonoids in plants [38]. *B. papyrifera* is a typical traditional Chinese medicine. Moreover, it is also a good feed material because of its high protein content. In short, *B. papyrifera* is not only a good feed material, but it also has important clinical medicinal value. So, it is of great significance to study the bioactive substances of *B. papyrifera*. In this work, we studied the polyphenol compounds in *B. papyrifera.* Flavonoid *C*-glycosides, among which there were only two aglycones, namely, apigenin and luteolin, and flavonoid *O*-glycosides were identified by the characteristics of UV-vis absorption spectroscopy combined with mass spectrum data (Table 1). There was no difference in the constitution of both flavonoid *O*-glycosides and flavonoid *C*-glycosides among the different samples. However, the content of the flavonoids varied widely among the different samples. Taking both the composition and content into consideration, hybrid paper mulberry could serve as another important natural source of flavonoid *C*-glycosides. This study showed that the content of chlorogenic acid and neo chlorogenic acid was very abundant in leaves. Chlorogenic acid, existing in natural plants such as *Lonicerae japonicae flos*, *Lonicerae flos* and *Eucommia ulmoides*, has a variety of biological activities including an anti-inflammatory property and an ability to prevent diseases. It can also be used as health medicine or as a food additive [39]. Previous studies have shown that the expression levels of critical inflammation molecules (interleukin-1β, interleukin-6, tumor necrosis factor-α, and nuclear factor-κB) were down-regulated in jejunal and ileal mucosa and the expression levels of inflammation repressors (suppressor of cytokine signaling 1 and toll-interacting protein) were up-regulated by chlorogenic acid [40]. Recent studies have shown that chlorogenic acid has a positive effect on improving the intestinal health of animals and enhancing the body’s antioxidant capacity, with great potential for application in livestock and poultry production [41]. Many phenolic compounds are the main natural antioxidants in plants, and previous studies have also suggested that the higher total phenolic compound content reflected a higher antioxidant activity. Our results demonstrated that the hybrid paper mulberry leaves possessed a high total flavonoid content and strong antioxidant activity. As a common medicine and edible homologous plant, hybrid paper mulberry contains alkaloids, flavonoid *C*-glycosides, flavonoid *O*-glycosides, vitamins and other chemical constituents. Our study showed that nine flavonoid *C*-glycosides (peaks 3, 4, 5, 7, 8, 9, 10, 11, 12) and five flavonoid *O*-glycosides (peaks 6, 13, 14, 15, 16) were detected in paper mulberry and hybrid paper mulberry. In particular, the content of orientin and vitexin was quite high. It was found that orientin and vitexin increased the antioxidant activity of serum and tissue and decreased the amount of malondialdehyde in mice [42]. Flavonoid *C*-glycosides and flavonoid *O*-glycosides appear to have positive influences on human health, and specifically have antioxidant, hepatoprotective, anticancer and antidiabetic potential [43]. On the basis of the known biosynthetic activities in higher plants and the compounds detected in paper mulberry, a possible flavonoid *C*- and *O*-glycosides biosynthesis pathway was proposed (Appendix A).

## 5. Conclusions

In this study, 155 *BpUGT* genes were identified in the *B. papyrifera* genome. These genes were clustered into 15 distinct evolutionary groups (A-O) based on the phylogenetic analysis. These different groups provide a useful foundation for understanding the structure-function relationships among the UGT family members. This work is the first to detect the functional characterization of the *UGT* gene family in paper mulberry. Moreover, it will provide novel insights into the functional analysis of the special traits of related gene families in plants. A total of 19 chemical compounds were identified and quantified by UPLC-ESI-MS/MS. Compared with *B. papyrifera* and *B. kazinoki*, the number of total phenols and flavonoids in the hybrid paper mulberry leaves was the highest. The phenol contents, flavonoid contents and antioxidant activities of leaves were determined using DPPH, FRAP and ABTS assays. All assays exhibited the same trend: the hybrid paper mulberry leaves showed a higher total flavonoid content, a higher total phenol content and greater antioxidant activities than the other leaves of *B. papyrifera* and *B. kazinoki*. Thus, the hybrid paper mulberry is the type of plant with development value, and the biological activity of its compounds needs to be further studied and exploited from the molecular level and the gene level.

## Figures and Tables

**Figure 1 molecules-26-03449-f001:**
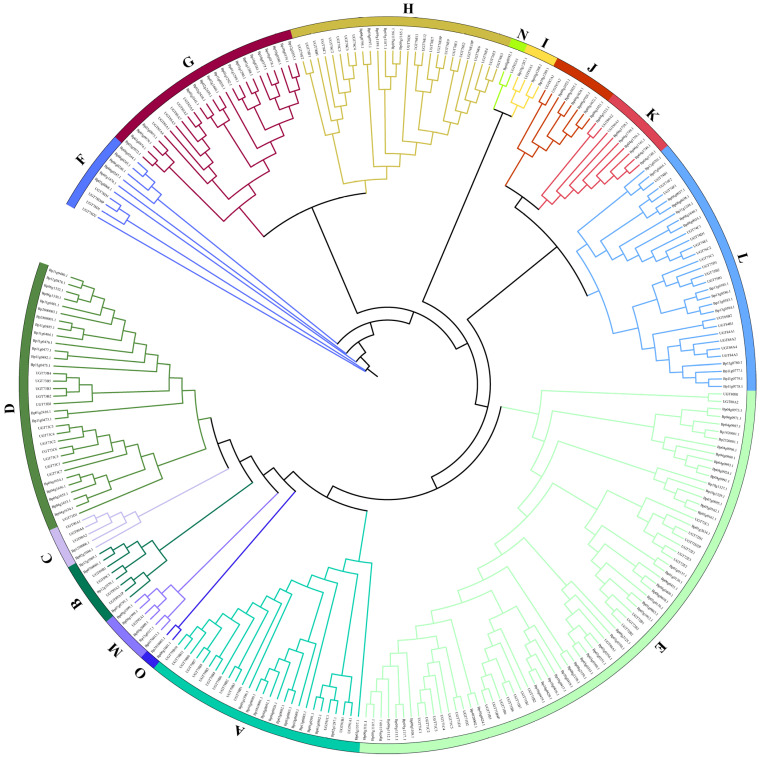
Phylogenetic analysis of the BpUGT proteins among *B. papyrifera* and *A. thaliana*. The full-length amino acid sequences of the BpUGT proteins from *B. papyrifera* and *A. thaliana* were aligned, and the phylogenetic tree was constructed by IQTREE using the maximum likelihood (ML) method with 1000 bootstrap replicates. The numbers on each branch line represent the bootstrap values. Different colored strips indicate subfamilies.

**Figure 2 molecules-26-03449-f002:**
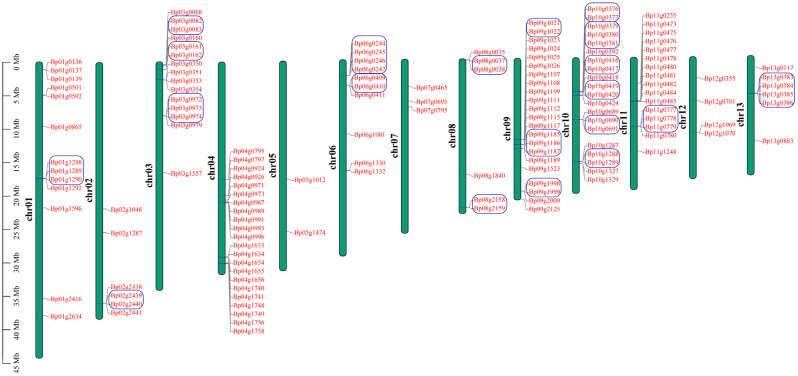
Chromosomal distribution and the tandem duplication of paper mulberry *BpUGT* genes. Green bars represent the chromosomes. Chromosome numbers are shown at the left of the bar. *BpUGT* genes are labeled at the right of the chromosomes. The tandem duplicated genes are represented by a blue borderline. The scale bar on the left indicates the chromosome lengths (Mb).

**Figure 3 molecules-26-03449-f003:**
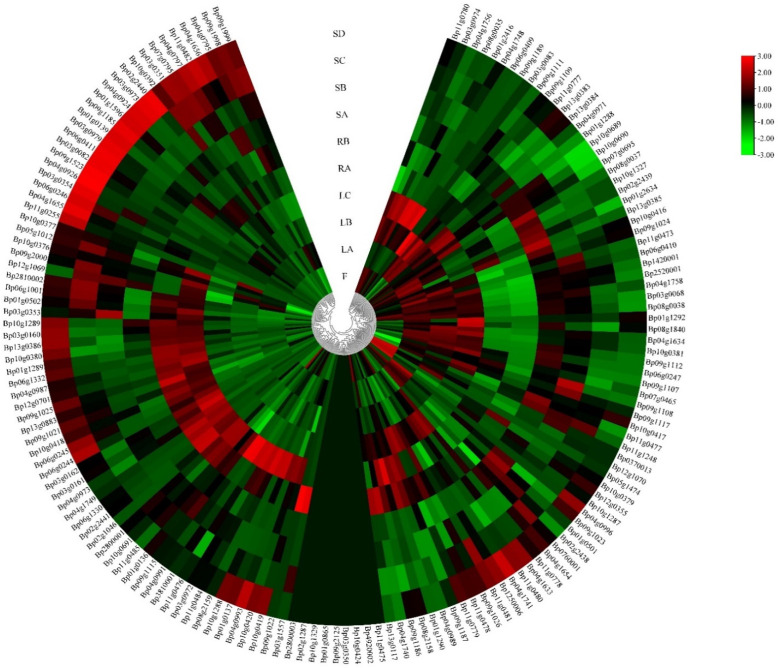
Expression profiles of the *BpUGT* genes with hierarchical clustering in different tissues. A cluster dendrogram is shown in the middle. The color scale at the right of the dendrogram represents log2-based expression values transformed from fragments per kilobase of exon per million reads mapped (FPKM) values. Red and green colors indicate higher levels and lower levels, respectively. SD: phloem of mature stem; SC: phloem of proximal stem; SB: immature stem; SA: shoot apex; RB: phloem of root; RA: root tip; LC: mature leaf; LB: developing leaf; LA: young leaf; F: fruit.

**Figure 4 molecules-26-03449-f004:**
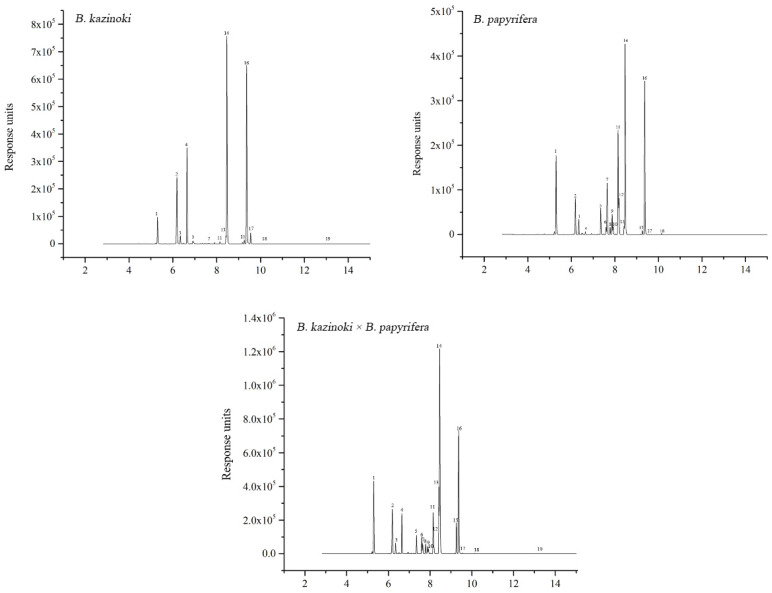
The UPLC elution profile of detected compounds from *B. papyrifera*, *B. kazinoki* and *B. kazinoki* × *B. papyrifera* leaves. The number of the peaks corresponds to that of the compounds in Table 1.

**Figure 5 molecules-26-03449-f005:**
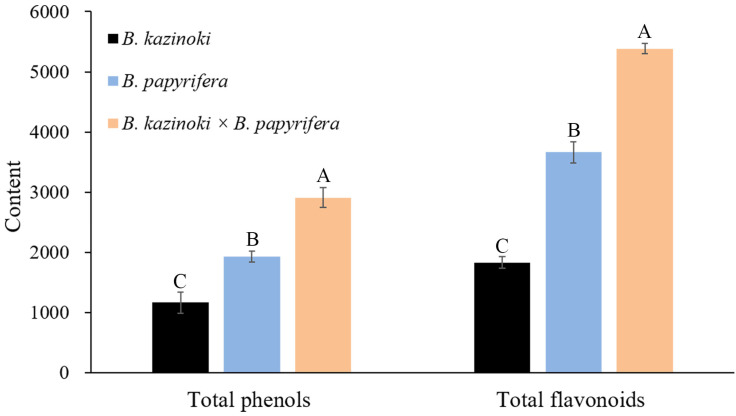
Total polyphenol content (TPC) and total flavonoid content (TFC) of leaf extracts from *B. kazinoki*, *B. papyrifera* and *B. kazinoki* × *B. papyrifera*. Letters above each pillar indicate significantly different values among samples according to ANOVA at *p* < 0.01. Values are mean ± SD (*n* = 3). Letters above each pillar indicate significantly different values among samples according to ANOVA at *p* < 0.01.

**Table 1 molecules-26-03449-t001:** The retention time, UV absorption maxima and MS data of the detected compounds from *B. kazinoki*, *B. papyrifera*, and *B. kazinoki* × *B. papyrifera* leaves.

Peak	RT ^a^ (min)	UV (nm)	[M]^+^ (*m*/*z*)	Identification	Reference
1	5.30	236, 325	355	Neochlorogenic acid	Standard
2	6.18	240, 326	355	Chlorogenic acid	Standard
3	6.19	270, 341	611	Di-*C*, *C*-hexosyl-luteolin	Hasegawa et al. (2008)
4	6.64	271, 332	595	Di-*C*, *C*-hexosyl-apigenin	Hasegawa et al. (2008)
5	7.35	269, 349	449, 433	Isoorientin	Standard
6	7.59	271, 337	595, 433	Vitexin-7-*O*-β-D-glucopyranoside	Feng et al. (2008)
7	7.64	269, 335	449, 433	Orientin	Standard
8	7.78	271, 337	565, 433	Schaftoside	Standard
9	7.88	270, 337	579, 433	5,7,4’-trihydroxyl-6-*C*-Rhamnopyranosyl-β-D-glucopyranosyl flavone	Feng et al. (2008)
10	7.92	270, 335	565, 433	Isoschaftoside	Standard
11	8.15	270, 337	433	Vitexin	Standard
12	8.16	271, 338	579, 433	5,7,4’-trihydroxyl-8-*C*-Rhamnopyranosyl-β-D-glucopyranosyl flavone	Feng et al. (2008)
13	8.46	272, 325	449, 287	Luteolin-7-*O*-β-D-glucopyranoside	Standard
14	8.47	254, 348	463, 287	Luteolin-7-*O*-β-D-glucopyranuronide	Yang et al. (2014)
15	9.26	267, 339	433, 271	Apigenin-7-*O*-β-D-glucopyranoside	Yang et al. (2014)
16	9.37	266, 337	447, 271	Apigenin-7-*O*-β-D-glucopyranuronide	Wang et al. (2011)
17	9.55	267, 347	477, 273	Dihydroapigenin derivative	Hasegawa et al. (2008)
18	11.83	253, 348	287	Luteolin	Standard
19	13.53	248, 346	271	Apigenin	Standard

^a^ RT: retention time on UPLC analysis.

**Table 2 molecules-26-03449-t002:** Antioxidant activity of *B. kazinoki*, *B. papyrifera* and *B. kazinoki* × *B. papyrifera* leaves.

Species	DPPH (Mg GAE/100 g DW)	FRAP (Mg GAE/100 g DW)	ABTS (Mg GAE/100 g DW)
*B. kazinoki*	530.5 ± 0.6 ^c^	329.5 ± 0.5 ^c^	195.6 ± 1.3 ^c^
*B. papyrifera*	606.8 ± 0.8 ^b^	384.0 ± 0.6 ^b^	214.5 ± 0.2 ^b^
*B. kazinoki × B. papyrifera*	700.2 ± 0.1 ^a^	448.3 ± 0.2 ^a^	233.0 ± 2.1 ^a^

Letters on the top right corner of each value indicate significantly different values among the samples according to ANOVA at *p* < 0.05. Values are mean ± SD (*n* = 3).

## Data Availability

Not applicable.

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
