# Peer review of "Genome-Wide Analysis of the UGT Gene Family and Identification of Flavonoids in Broussonetia papyrifera"

_molecules, 2021, doi:10.3390/molecules26113449_

Round 1
Reviewer 1 Report
The authors characterized the UGT from B. papyrifera genome using a classical genome-wide approach.
They also characterized the phytochemistry targeted to flavonoids mainly of 3 Broussonetia species (ie: B. papyrifera, B. kazinoki, and hybrid paper mulberry). 1) Please add a table or figure showing the relative-quantitative and qualitative differences in accumulation profiles of these 3 species.
Then the authors compared the antioxidant capacity of the 3 species using in vitro assays. 2) it is unclear how the authors have chosen these assays among the existing multitude of antioxidant assays.
My main criticism is about the absence of clear links between these 3 different parts. The authors should improve this point. Here, we can consider these results as at least 2 separated studies without strong links.
Minor changes are also required:
Cis-regulatory analysis: “cis” in italics and after in the text
“with 1000 bootstrap replicates”: “1,000” not “1000” same remark for other numbers in the text
Precise injection volumes for UPLC-PDA and UPLC-MS/MS conditions
Quality of Figure 1 is low for publication, protein names cannot be read.
Similarly improve Figure 2 quality
Names are difficult to read in Figure 3 too
Author Response
Please see the attacment.

Reviewer 2 Report
I consider that the article can be published in this form, it is very interesting for many people from different fields of activity.
These are my detailed comments:1. Authors provided a paper entitled “Genome-wide analysis of the UGT gene family and identification of flavonoids in Broussonetia papyrifera” for the publication on Molecules, MDPI.
2. This work is the first to detect the functional characterization of UGT gene family in paper mulberry. A total of 19 chemical compounds were identified and quantified. The hybrid paper mulberry leaves (B. kazinoki x B. papyrifera) showed higher TFC, TPC and antioxidant activities than the leaves of B. papyrifera and B. kazinoki. This makes the hybrid paper mulberry is a plant with possible therapeutic value.
There are some typing errors (papyifera, instead papyrifera), so maybe English should be revised.This paper has a good scientific soundness and deserves to be published.
Author Response
Please see the attacment.

Reviewer 3 Report
In its present form, the manuscript requires some corrections and additions to improve its great value. My suggestions below:
- Please, make the abstract contain more information on the obtained research results.
-
What purity class was the standards used?
- In the article 'Isolation, purification and antioxidant activity of polysaccharides from the leaves of maca (Lepidium Meyeni)', the DPPH radical scavenging activity was measured at a wavelength of 517 nm. This article was based on the cited methodology which also used the same wavelength. Please, also cite other studies in which the 515 nm wavelength was used. I suspect that the choice of this wavelength was a consequence of your research.
- Please include more comparative analysis on antioxidant properties in the scientific discussion.
- If possible, please supplemented scientific discussion with the currently published, current scientific works. Unfortunately, most of the articles used are not from recent years.
- The abstract contained the sentence 'Given its high total flavonoid content and strong antioxidant activity, B. papyifera × B. kazinoki may be a potential functional health food'. Please expand the topic of the possibility of using the plant as functional food by explainig this term and adding examples of the same or similar applications already present.
Reviewer 4 Report
In the paper entitled "Genome-wide analysis of the UGT gene family and identification of flavonoids in Broussonetia papyrifera", 155 BpUGT genes were identified in the B. papyrifera genome and a total of 19 chemical compounds were identified and quantified by UPLC-ESI-MS/MS consisting in polyphenols and flavonoids. The phenol contents, flavonoid contents and antioxidant activities of 3 species of Broussonetia were analyzed. This study provides valuable information for understanding the function of BpUGTs in the biosynthesis of flavonoids.
The paper is interesting, the methodology is adequate and explicitly stated, the style is clear and concise and the subject is very topical. For this reason, I recommend the publication of this study.
Reviewer 5 Report
The manuscript by Fenfen et al. uses bioinformatic, biochemical, and molecular approaches to systematic identify and analysis the UGT gene family in Broussonetia papyrifera. The authors have generated a solid amount of data and the conclusions are justified by the results. However, some details are missing or need to be clarified in the text and legends.
The manuscript should have the line numbers for reviewing.
Page 7: the legends of Figure 1: “Different species are shown in different colored fonts.” is not clear.
Page 7: Remove “There were 46 UGT members in the largest E group and only one UGT member in the smallest N group.”. It has mentioned before.
Page 8: the legends of Figure 2: “The tandem duplicated genes are represented by red borderline, and the gene clusters are boxed together by blue lines.”. Figure 2 has red boxes but no blue lines, please clarify.
Page 10: the legends of Figure 3: please indicate the full names of the abbreviation “SD, SC, SB, SA, RB, RA, LC, LB, and F”.
Page 11: the legends of Figure 4: please mention the numbers on the top of peaks correspond to the compounds in Table 1.
Page 11: Table 1: Add another column or clarify that which 18, 13, 19 compounds are identified from B. papyrifera, B. kazinoki, and hybrid paper mulberry, respectively.
Page 14: Figure 5: It is better to rearrange the results with two groups, total phenols group (dark grey bars) and total flavonoids group (light grey bars).
Page 14: the legends of Figure 5: mention error bars for SD? or others, and the meaning of letters on the top of columns.
Round 2
Reviewer 1 Report
Revision has been performed adequately.
All my concerns have been successfully considered by the Authors.